# Economy Statistical Recurrent Units For Inferring Nonlinear Granger Causality

**Saurabh Khanna**
Department of Electrical and Computer Engineering
National University of Singapore
elesaur@nus.edu.sg

**Vincent Y. F. Tan**
Department of Electrical and Computer Engineering
Department of Mathematics
National University of Singapore
vtan@nus.edu.sg

## Abstract

Granger causality is a widely-used criterion for analyzing interactions in large-scale networks. As most physical interactions are inherently nonlinear, we consider the problem of inferring the existence of pairwise Granger causality between nonlinearly interacting stochastic processes from their time series measurements. Our proposed approach relies on modeling the embedded nonlinearities in the measurements using a component-wise time series prediction model based on Statistical Recurrent Units (SRUs). We make a case that the network topology of Granger causal relations is directly inferrable from a structured sparse estimate of the internal parameters of the SRU networks trained to predict the processes' time series measurements. We propose a variant of SRU, called *economy-SRU*, which, by design has considerably fewer trainable parameters, and therefore less prone to overfitting. The economy-SRU computes a low-dimensional sketch of its high-dimensional hidden state in the form of random projections to generate the feedback for its recurrent processing. Additionally, the internal weight parameters of the economy-SRU are strategically regularized in a group-wise manner to facilitate the proposed network in extracting meaningful predictive features that are highly time-localized to mimic real-world causal events. Extensive experiments are carried out to demonstrate that the proposed economy-SRU based time series prediction model outperforms the MLP, LSTM and attention-gated CNN-based time series models considered previously for inferring Granger causality.

## 1 Introduction

The physical mechanisms behind the functioning of any large-scale system can be understood in terms of the networked interactions between the underlying system processes. Granger causality is one widely-accepted criterion used in building network models of interactions between large ensembles of stochastic processes. While Granger causality may not necessarily imply true causality, it has proven effective in qualifying pairwise interactions between stochastic processes in a variety of system identification problems, e.g., gene regulatory network mapping (Fujita et al. (2007)), and the mapping of human brain connectome (Seth et al. (2015)). This perspective has given rise to the canonical problem of inferring pairwise Granger causal relationships between a set of stochastic processes from their time series measurements. At present, the vast majority of Granger causal inference methods adopt a model-based inference approach whereby the measured time series data is modeled using with a suitable parameterized data generative model whose inferred parameters ultimately reveal the true topology of pairwise Granger causal relationships. Such methods typically rely on using linear regression models for inference. However, as illustrated in the classical bivariate example by Baek & Brock (1992), linear model-based Granger causality tests can fail catastrophically in the presence of even mild nonlinearities in the measurements, thus making a strong case for our work which tackles the nonlinearities in the measurements by exploring new generative models of the time series measurements based on recurrent neural networks.

## 2 PROBLEM FORMULATION

Consider a multivariate dynamical system whose evolution from an initial state is fully characterized by $n$ distinct stochastic processes which can potentially interact nonlinearly among themselves. Our goal here is to unravel the unknown nonlinear system dynamics by mapping the entire network of pairwise interactions between the system-defining stochastic processes, using Granger causality as the qualifier of the individual pairwise interactions.

In order to detect the pairwise Granger causal relations between the stochastic processes, we assume access to their concurrent, uniformly-sampled measurements presented as an $n$-variate time series $\mathbf{x} = \{\mathbf{x}_t : t \in \mathbb{N}\} \subset \mathbb{R}^n$. Let $\mathbf{x}_{t,i}$ denote the $i^{\text{th}}$ component of the $n$-dimensional vector measurement $\mathbf{x}_t$, representing the measured value of process $i$ at time $t$. Motivated by the framework proposed in Tank et al. (2017), we assume that the measurement samples $\mathbf{x}_t, t \in \mathbb{N}$ are generated sequentially according to the following nonlinear, component-wise autoregressive model:

$$\mathbf{x}_{t,i} = f_i\left(\mathbf{x}_{t-p:t-1,1}, \mathbf{x}_{t-p:t-1,2}, \ldots, \mathbf{x}_{t-p:t-1,n}\right) + \mathbf{e}_{t,i}, \quad i = 1, 2, \ldots n, \tag{1}$$

where $\mathbf{x}_{t-p:t-1,j} \triangleq \{\mathbf{x}_{t-1,j}, \mathbf{x}_{t-2,j}, \ldots, \mathbf{x}_{t-p,j}\}$ represents the most recent $p$ measurements of the $j^{\text{th}}$ component of $\mathbf{x}$ in the immediate past relative to current time $t$. The scalar-valued component generative function $f_i$ captures all of the linear and nonlinear interactions between the $n$ stochastic processes up to time $t - 1$ that decide the measured value of the $i^{\text{th}}$ stochastic process at time $t$. The residual $\mathbf{e}_{i,t}$ encapsulates the combined effect of all instantaneous and exogenous factors influencing the measurement of process $i$ at time $t$, as well as any imperfections in the presumed model. Equation 1 may be viewed as a generalization of the linear vector autoregressive (VAR) model in the sense that the components of $\mathbf{x}$ can be nonlinearly dependent on one another across time. The value $p$ is loosely interpreted to be the *order* of the above nonlinear autoregressive model.

### 2.1 GRANGER CAUSALITY IN NONLINEAR DYNAMICAL SYSTEMS

We now proceed to interpret Granger causality in the context of the above component-wise time series model. Recalling the standard definition by Granger (1969), a time series $\mathbf{v}$ is said to *Granger cause* another time series $\mathbf{u}$ if the past of $\mathbf{v}$ contains new information above and beyond the past of $\mathbf{u}$ that can improve the predictions of current or future values of $\mathbf{u}$. For $\mathbf{x}$ with its $n$ components generated according to equation 1, the concept of Granger causality can be extended as suggested by Tank et al. (2018) as follows. We say that series $j$ *does not Granger cause* series $i$ if the component-wise generative function $f_i$ does not depend on the past measurements in series $j$, i.e., for all $t \geq 1$ and all distinct pairs $\mathbf{x}_{t-p:t-1,j}$ and $\mathbf{x}'_{t-p:t-1,j}$,

$$f_i\left(\mathbf{x}_{t-p:t-1,1}, \ldots, \mathbf{x}_{t-p:t-1,j}, \ldots, \mathbf{x}_{t-p:t-1,n}\right) = f_i\left(\mathbf{x}_{t-p:t-1,1}, \ldots, \mathbf{x}'_{t-p:t-1,j}, \ldots, \mathbf{x}_{t-p:t-1,n}\right). \tag{2}$$

From equation 1, it is immediately evident that under the constraint in equation 2, the past of series $j$ does not assert any causal influence on series $i$, in alignment with the core principle behind Granger causality. Based on the above implication of equation 2, the detection of Granger noncausality between the components of $\mathbf{x}$ translates to identifying those components of $\mathbf{x}$ whose past is irrelevant to the functional description of each individual $f_i$ featured in equation 1.

Note that any reliable inference of pairwise Granger causality between the components of $\mathbf{x}$ is feasible only if there are no unobserved confounding factors in the system which could potentially influence $\mathbf{x}$. In this work, we assume that the system of interest is *causally sufficient* (Spirtes & Zhang (2016)), i.e., none of the $n$ stochastic processes (whose measurements are available) have a common Granger-causing-ancestor that is unobserved.

### 2.2 INFERRING GRANGER CAUSALITY USING COMPONENT-WISE RECURRENT MODELS

We undertake a model-based inference approach wherein the time series measurements are used as observations to learn an autoregressive model which is anatomically similar to the component-wise generative model described in equation 1 except for the unknown functions $f_i$ replaced with their respective parameterized approximations denoted by $g_i$. Let $\Theta_i, 1 \leq i \leq n$ denote the complete set of parameters encoding the functional description of the approximating functions $\{g_i\}_{i=1}^n$. Then, the pairwise Granger causality between series $i$ and the components of $\mathbf{x}$ is deduced from $\Theta_i$ which is estimated by fitting $g_i$'s output to the ordered measurements in series $i$. Specifically, if the estimated $\Theta_i$ suggests that $g_i$'s output is independent of the past measurements in series $j$, then

we declare that series $j$ is Granger noncausal for series $i$. We aim to design the approximation function $g_i$ to be highly expressive and capable of well-approximating any intricate causal coupling between the components of $\mathbf{x}$ induced by the component-wise function $f_i$, while simultaneously being easily identifiable from underdetermined measurements.

By virtue of their universal approximation property (Schäfer & Zimmermann (2006)), recurrent neural networks or RNNs are a particularly ideal choice for $g_i$ towards inferring the pairwise Granger causal relationships in $\mathbf{x}$. In this work, we investigate the use of a special type of RNN called the *statistical recurrent unit* (SRU) for inferring pairwise Granger causality between multiple nonlinearly interacting stochastic processes. Introduced by Oliva et al. (2017), an SRU is a highly expressive recurrent neural network designed specifically for modeling multivariate time series data with complex-nonlinear dependencies spanning multiple time lags. Unlike the popular gated RNNs (e.g., long short-term memory (LSTM) (Hochreiter & Schmidhuber (1997)) and gated recurrent unit (GRU)) (Chung et al. (2014)), the SRU's design is completely devoid of the highly nonlinear sigmoid gating functions and thus less affected by the vanishing/exploding gradient issue during training. Despite its simpler ungated architecture, an SRU can model both short and long-term temporal dependencies in a multivariate time series. It does so by maintaining multi-time scale summary statistics of the time series data in the past, which are preferentially sensitive to different older portions of the time series $\mathbf{x}$. By taking appropriate linear combinations of the summary statistics at different time scales, an SRU is able to construct predictive causal features which can be both highly component-specific and lag-specific at the same time. From the causal inference perspective, this dual-specificity of the SRU's predictive features is its most desirable feature, as one would argue that causal effects in reality also tend to be highly localized in both space and time.

The main contributions of this paper can be summarized as follows:

1. We propose the use of statistical recurrent units (SRUs) for detecting pairwise Granger causality between the nonlinearly interacting stochastic processes. We show that the entire network of pairwise Granger causal relationships can be inferred directly from the regularized block-sparse estimate of the input-layer weight parameters of the SRUs trained to predict the time series measurements of the individual processes.

2. We propose a modified SRU architecture called *economy SRU* or *eSRU* in short. The first of the two proposed modifications is aimed at substantially reducing the number of trainable parameters in the standard SRU model without sacrificing its expressiveness. The second modification entails regularizing the SRU's internal weight parameters to enhance the interpretability of its learned predictive features. Compared to the standard SRU, the proposed eSRU model is considerably less likely to overfit the time series measurements.

3. We conduct extensive numerical experiments to demonstrate that eSRU is a compelling model for inferring pairwise Granger causality. The proposed model is found to outperform the multi-layer perceptron (MLP), LSTM and attention-gated convolutional neural network (AG-CNN) based models considered in the earlier works.

## 3 Proposed Granger causal inference framework

In the proposed scheme, each of the unknown generative functions $f_i, 1 \leq i \leq n$ in the presumed component-wise model of $\mathbf{x}$ in (1) is individually approximated by a distinct SRU network. The $i^{\text{th}}$ SRU network sequentially processes the time series measurements $\mathbf{x}$ and outputs a next-step prediction sequence $\hat{\mathbf{x}}_i^+ = \{\hat{x}_{i,2}, \hat{x}_{i,3}, \ldots, \hat{x}_{i,t+1}, \ldots\} \subset \mathbb{R}$, where $\hat{x}_{i,t+1}$ denotes the predicted value of component series $i$ at time $t + 1$. The prediction $\hat{x}_{i,t+1}$ is computed in a recurrent fashion by combining the current input sample $\mathbf{x}_t$ at time $t$ with the summary statistics of past samples of $\mathbf{x}$ up to and including time $t - 1$ as illustrated in Figure 1.

The following update equations describe the sequential processing of the input time series $\mathbf{x}$ within the $i^{\text{th}}$ SRU network in order to generate a prediction of $x_{i,t+1}$.

$$\text{Feedback: } \mathbf{r}_{i,t} = h\left(\boldsymbol{W}_{\text{r}}^{(i)}\mathbf{u}_{i,t-1} + \boldsymbol{b}_{\text{r}}^{(i)}\right) \in \mathbb{R}^{d_{\text{r}}}. \tag{3a}$$

$$\text{Recurrent statistics: } \phi_{i,t} = h\left(\boldsymbol{W}_{\text{in}}^{(i)}\mathbf{x}_t + \boldsymbol{W}_{\text{f}}^{(i)}\mathbf{r}_{i,t-1} + \boldsymbol{b}_{\text{in}}^{(i)}\right) \in \mathbb{R}^{d_{\phi}}. \tag{3b}$$

$$\text{Multi-scale summary statistics: } \mathbf{u}_{i,t} = \left[ \left( \mathbf{u}_{i,t}^{\alpha_1} \right)^T \left( \mathbf{u}_{i,t}^{\alpha_2} \right)^T \ldots \left( \mathbf{u}_{i,t}^{\alpha_m} \right)^T \right]^T \in \mathbb{R}^{md_\phi}, \alpha_j \in \mathcal{A}, \forall j. \tag{3c}$$

$$\text{Single-scale summary statistics: } \mathbf{u}_{i,t}^{\alpha_j} = (1 - \alpha_j)\mathbf{u}_{i,t-1}^{\alpha_j} + \alpha_j\phi_{i,t}, \ \in \mathbb{R}^{d_\phi}, \alpha_j \in [0,1]. \tag{3d}$$

$$\text{Output features: } \mathbf{o}_{i,t} = h\left( \boldsymbol{W}_{\mathrm{o}}^{(i)}\mathbf{u}_{i,t} + \boldsymbol{b}_{\mathrm{o}}^{(i)} \right) \in \mathbb{R}^{d_\mathrm{o}}. \tag{3e}$$

$$\text{Output prediction: } \hat{\mathrm{x}}_{i,t+1} = \left( \boldsymbol{w}_{\mathrm{y}}^{(i)} \right)^T \mathbf{o}_{i,t} + b_{\mathrm{y}}^{(i)} \in \mathbb{R}. \tag{3f}$$

The function $h$ in the above updates is the elementwise Rectified Linear Unit (ReLU) operator, $h(\cdot) := \max(\cdot, 0)$, which serves as the nonlinear activation in the three dedicated single layer neural networks that generate the recurrent statistics $\phi_{i,t}$, the feedback $\mathbf{r}_{i,t}$ and the output features $\mathbf{o}_{i,t}$ in the $i^{\mathrm{th}}$ SRU network. In order to generate the next-step prediction of series $i$ at time $t$, the $i^{\mathrm{th}}$ SRU network first prepares the feedback $\mathbf{r}_{i,t}$ by nonlinearly transforming its last hidden state $\mathbf{u}_{i,t-1}$. As stated in equation 3a, a single layer ReLU network parameterized by weight matrix $\boldsymbol{W}_{\mathrm{r}}^{(i)}$ and bias $\boldsymbol{b}_{\mathrm{r}}^{(i)}$ maps the hidden state $\mathbf{u}_{i,t-1}$ to the feedback $\mathbf{r}_{i,t}$. Another single layer ReLU network parameterized by weight matrices $\boldsymbol{W}_{\mathrm{in}}^{(i)}, \boldsymbol{W}_{\mathrm{f}}^{(i)}$ and bias $\boldsymbol{b}_{\mathrm{in}}^{(i)}$ takes the input $\mathbf{x}_t$ and the feedback $\mathbf{r}_{i,t}$ and tranforms them into the recurrent statistics $\phi_{i,t}$ as described in equation 3b. Equation 3d describes how the network's multi-timescale hidden states $\mathbf{u}_{i,t}^\alpha$ for $\alpha \in \mathcal{A} = \{\alpha_1, \alpha_2, \ldots, \alpha_m\} \subset [0,1]$ are updated in parallel by taking exponentially weighted moving averages of the recurrent statistics $\phi_{i,t}$ corresponding to $m$ different scales in $\mathcal{A}$. A third single layer ReLU network parameterized by $\boldsymbol{W}_{\mathrm{o}}^{(i)}$ and $\boldsymbol{b}_{\mathrm{o}}^{(i)}$ transforms the concatenated multi-timescale summary statistics $\mathbf{u}_{i,t} = \left[ (\mathbf{u}_{i,t}^{\alpha_1})^T(\mathbf{u}_{i,t}^{\alpha_2})^T \ldots (\mathbf{u}_{i,m}^{\alpha_m})^T \right]^T$ to generate the nonlinear causal features $\mathbf{o}_{i,t}$ which, according to Oliva et al. (2017), are arguably highly sensitive to the input time series measurements at specific lags. Finally, the network generates the next-step prediction of series $i$ as $\hat{\mathrm{x}}_{i,t+1}$ by linearly combining the nonlinear output features in $\mathbf{o}_{i,t}$, as depicted in equation 3f.

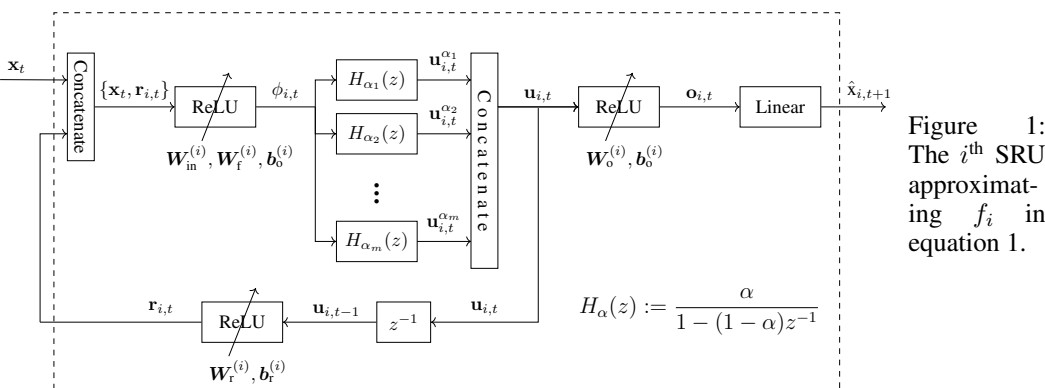

Figure 1: The $i^{\mathrm{th}}$ SRU approximating $f_i$ in equation 1.

For values of scale $\alpha \approx 1$, the single-scale summary statistic $\mathbf{u}_{i,t}^\alpha$ in equation 3d is more sensitive to the recent past measurements in $\mathbf{x}$. On the other hand, $\alpha \approx 0$ yields a summary statistic that is more representative of the older portions of the input time series. Oliva et al. (2017) elaborates on how the SRU is able to generate output features $(\mathbf{o}_{i,t}, 1 \leq i \leq n)$ that are preferentially sensitive to the measurements from specific past segments of $\mathbf{x}$ by taking appropriate linear combinations of the summary statistics corresponding to different values of $\alpha$ in $\mathcal{A}$.

### 3.1 INFERRING PAIRWISE GRANGER CAUSALITY USING SRUs

Let $\Theta_{\mathrm{SRU}}^{(i)} \triangleq \{ \boldsymbol{W}_{\mathrm{f}}^{(i)}, \boldsymbol{W}_{\mathrm{in}}^{(i)}, \boldsymbol{b}_{\mathrm{in}}^{(i)}, \boldsymbol{W}_{\mathrm{r}}^{(i)}, \boldsymbol{b}_{\mathrm{r}}^{(i)}, \boldsymbol{W}_{\mathrm{o}}^{(i)}, \boldsymbol{b}_{\mathrm{o}}^{(i)}, \boldsymbol{w}_{\mathrm{y}}^{(i)}, b_{\mathrm{y}}^{(i)} \}$ denote the complete set of parameters of the $i^{\mathrm{th}}$ SRU network approximating $f_i$ in the presumed component-wise model of $\mathbf{x}$. From equation 3b, we observe that the weight matrix $\boldsymbol{W}_{\mathrm{in}}^{(i)}$ regulates the influence of the individual components of the input time series $\mathbf{x}$ on the generation of the recurrent statistics $\phi_{i,t}$, and ultimately the next-step prediction of series $i$. In real-world dynamical systems, the networked interactions are typically sparse which implies that very few dimensions of the input time series $\mathbf{x}$ actually play a role in the generation of its individual components. Bearing this property of the networked interactions in mind, we are interested in learning the parameters $\Theta_{\mathrm{SRU}}^{(i)}$ such that the $i^{\mathrm{th}}$ SRU's sequential

output closely matches with series $i$'s measurements, while simultaneously seeking a *column-sparse* estimate of the weight matrix $\boldsymbol{W}_{\text{in}}^{(i)}$.

We propose to learn the parameters $\Theta_{\text{SRU}}^{(i)}$ of the $i^{\text{th}}$ SRU network by minimizing the penalized mean squared prediction error loss as shown below.

$$\hat{\Theta}_{\text{SRU}}^{(i)} := \arg\min_{\Theta_{\text{SRU}}^{(i)}} \frac{1}{T-1} \sum_{t=1}^{T-1} (\hat{\text{x}}_{i,t} - \text{x}_{i,t+1})^2 + \lambda_1 \sum_{j=1}^{n} \|\boldsymbol{W}_{\text{in}}^{(i)}(:,j)\|_2. \tag{4}$$

In the above, the network output $\hat{\text{x}}_{i,t}$ depends nonlinearly on $\boldsymbol{W}_{\text{in}}^{(i)}$ according to the composite relation described by the updates (3a)-(3f) and $\boldsymbol{W}_{\text{in}}^{(i)}(:,j)$ denotes the $j^{\text{th}}$ column in the weight matrix $\boldsymbol{W}_{\text{in}}^{(i)}$. The $\ell_1$-group norm penalty in the objective is known to promote column sparsity in the estimated $\boldsymbol{W}_{\text{in}}^{(i)}$ (Simon et al. (2013)). From equation 3b, a straightforward implication of the column vector $\boldsymbol{W}_{\text{in}}^{(i)}(:,j)$ being estimated as the all-zeros vector is that the past measurements in series $j$ do not influence the predicted future value of series $i$. In this case, we declare that series $j$ does not Granger-cause series $i$. Moreover, the index set supporting the non-zero columns in the estimated weight matrix $\hat{\boldsymbol{W}}_{\text{in}}^{(i)}$ enumerates the components of $\mathbf{x}$ which are likely to Granger-cause series $i$. Likewise, the entire network of pairwise Granger causal relationships in $\mathbf{x}$ can be deduced from the non-zero column support of the estimated weight matrices $\boldsymbol{W}_{\text{in}}^{(i)}, 1 \leq i \leq n$ in the $n$ SRU networks trained to predict the components of $\mathbf{x}$.

The component-wise SRU optimization problem in equation 4 is nonconvex and potentially has multiple local minima. To solve for $\hat{\Theta}_{\text{SRU}}^{(i)}$, we use first-order gradient-based methods such as stochastic gradient descent which have been found to be consistently successful in finding good solutions of nonconvex deep neural network optimization problems (Allen-Zhu et al. (2019)). Since our approach of detecting Granger noncausality hinges upon correctly identifying the all-zero columns of $\boldsymbol{W}_{\text{in}}^{(i)}$, it is important that the first-order gradient based parameter updates used for minimizing the penalized SRU loss ensure that majority of the coefficients in $\boldsymbol{W}_{\text{in}}^{(i)}$ iterates become exactly zero after a certain number of iterations. Seeking exact column sparsity in the converged solution of $\boldsymbol{W}_{\text{in}}^{(i)}$, we follow the same approach as Tank et al. (2018) and resort to a first-order proximal gradient descent algorithm to find a regularized solution of the SRU optimization. The gradients needed for executing the gradient descent updates of the SRU network parameters are computed efficiently using the *backpropagation through time* (BPTT) procedure (Jaeger (2002)).

## 4 ECONOMY SRU: A REMEDY FOR OVERFITTING

By computing the summary statistics of past measurements at sufficiently granular time scales, an SRU can learn predictive causal features which are highly localized in time. While a higher granularity of $\alpha$ in $\mathcal{A}$ translates to a more general SRU model that fits better to the time series measurements, it also entails substantial increase in the number of trainable parameters. Since measurement scarcity is typical in causal inference problems, the proposed component-wise SRU based time series prediction model is usually overparameterized and thus susceptible to overfitting. The typical high dimensionality of the recurrent statistic $\phi_t$ accentuates this issue.

To alleviate the overfitting concerns, we propose two modifications to the standard SRU (Oliva et al. (2017)) aimed primarily at reducing its likelihood of overfitting the time series measurements. The modifications are relevant regardless of the current Granger causal inference context, and henceforth we refer to the modified SRU as *Economy-SRU (eSRU)*.

### 4.1 MODIFICATION-I: GENERATING FEEDBACK FROM A LOW-DIMENSIONAL SKETCH OF SUMMARY STATISTICS

We propose to reduce the number of trainable parameters in the $i^{\text{th}}$ SRU network by substituting the feedback ReLU network parameterized by $\boldsymbol{W}_{\text{r}}^{(i)}$ and $\boldsymbol{b}_{\text{r}}^{(i)}$ with the two stage network shown in Fig. 2. The first stage implements the linear matrix-vector multiplication operation $\boldsymbol{D}_{\text{r}}^{(i)}\mathbf{u}_{i,t}$ to generate the output $\boldsymbol{v}_{i,t} \in \mathbb{R}^{d_{\text{r}}'}$, where $\boldsymbol{D}_{\text{r}}^{(i)} \in \mathbb{R}^{d_{\text{r}}' \times md_\phi}$ is a fixed, full row-rank

matrix with $d'_{\mathrm{r}} \ll md_\phi$. The $d'_{\mathrm{r}}$-dimensional output of the first stage can be viewed as a low-dimensional, stable embedding of the multi-timescale summary statistics $\mathbf{u}_{i,t}$. The entries of the constant matrix $\boldsymbol{D}_{\mathrm{r}}^{(i)}$ are drawn independently from a zero mean Gaussian distribution with variance $\frac{1}{d'_{\mathrm{r}}}$. The stage-1 processing is based on the premise that for most real-world systems and

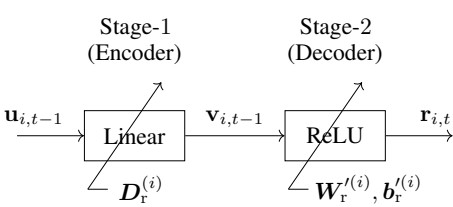

Figure 2: Proposed two-stage feedback in economy-SRU.

the associated time series measurements, their high-dimensional summary statistics learned by the SRU network as $\mathbf{u}_{i,t}$ tend to be highly structured, and thus $\mathbf{u}_{i,t}$ has significantly fewer degrees of freedom relative to its ambient dimension. Thus, by projecting the $md_\phi$-dimensional $\mathbf{u}_{i,t}$ onto the $d'_{\mathrm{r}} (\ll md_\phi)$ rows of $\boldsymbol{D}_{\mathrm{r}}^{(i)}$, we obtain its low-dimensional embedding $\boldsymbol{v}_{i,t}$ which nonetheless retains most of the contextual information conveyed by the uncompressed $\mathbf{u}_{i,t}$[1] Johnson & Lindenstrauss (1984); Dirksen (2014). The second stage of the proposed feedback network is a single/multi-layer ReLU network which maps the sketched summary statistics $\boldsymbol{v}_{i,t}$ to the feedback vector $\mathbf{r}_{i,t}$. The second stage ReLU network is parameterized by weight matrix $\boldsymbol{W}_{\mathrm{r}}^{\prime,(i)} \in \mathbb{R}^{d_{\mathrm{r}} \times d'_{\mathrm{r}}}$ and bias $\boldsymbol{b}_{\mathrm{r}}^{\prime,(i)} \in \mathbb{R}^{d_{\mathrm{r}}}$. Compared to the standard SRU's feedback whose generation is controlled by $md_\phi d_{\mathrm{r}} + d_{\mathrm{r}}$ trainable parameters, the proposed feedback network has only $d'_{\mathrm{r}} d_{\mathrm{r}} + d_{\mathrm{r}}$ trainable parameters, which is substantially fewer when $d'_{\mathrm{r}} \ll md_\phi$. Consequently, the modified SRU is less susceptible to overfitting.

## 4.2 MODIFICATION-II: GROUPED-SPARSE MIXING OF MULTI-SCALE SUMMARY STATISTICS FOR LEARNING TIME-LOCALIZED PREDICTIVE FEATURES

In the standard SRU proposed by Oliva et al. (2017), there are no restrictions on the weight matrix $\boldsymbol{W}_{\mathrm{o}}^{(i)}$ parameterizing the ReLU network that maps the summary statistics $\mathbf{u}_{i,t}$ to the final predictive features in $\mathbf{o}_{i,t}$. Noting that the number of parameters in the $md_\phi \times d_{\mathrm{o}}$ sized weight matrix $\boldsymbol{W}_{\mathrm{o}}^{(i)}$ usually dominates the overall number of trainable parameters in the SRU, any meaningful effort towards addressing the model overfitting concerns must consider regularizing the weights in $\boldsymbol{W}_{\mathrm{o}}^{(i)}$. In this spirit, we propose the following penalized optimization problem to estimate the parameters $\Theta_{\mathrm{eSRU}}^{(i)} = (\Theta_{\mathrm{SRU}}^{(i)} \backslash \{\boldsymbol{W}_{\mathrm{r}}^{(i)}\}) \cup \{\boldsymbol{W}_{\mathrm{r}}^{\prime(i)}\}$ of the eSRU model equipped with the two-stage feedback proposed in Section 4.1:

$$\hat{\Theta}_{\mathrm{eSRU}}^{(i)} = \underset{\Theta_{\mathrm{eSRU}}^{(i)}}{\arg\min} \frac{1}{T-1} \sum_{t=1}^{T-1} (\hat{\mathrm{x}}_{i,t} - \mathrm{x}_{i,t+1})^2 + \lambda_1 \sum_{j=1}^{n} \|\boldsymbol{W}_{\mathrm{in}}^{(i)}(:,j)\|_2 + \lambda_2 \sum_{j=1}^{d_{\mathrm{o}}} \sum_{k=1}^{d_\phi} \|\boldsymbol{W}_{\mathrm{o}}^{(i)}(j, \mathcal{G}_{j,k})\|_2.$$
(5)

Here $\lambda_1$ and $\lambda_2$ are positive constants that bias the group sparse penalizations against the eSRU's fit to the measurements in the $i^{\mathrm{th}}$ component series. The term $\boldsymbol{W}_{\mathrm{o}}^{i}(j, \mathcal{G}_{j,k})$ $(1 \le j \le d_{\mathrm{o}}, 1 \le k \le d_\phi)$ denotes the subvector of the $j^{\mathrm{th}}$ row in $\boldsymbol{W}_{\mathrm{o}}^{(i)}$ obtained by extracting the weight coefficients indexed by set $\mathcal{G}_{j,k}$. As shown via an example in Fig. 3, the index set $\mathcal{G}_{j,k}$ enumerates the $m$ weight coefficients in the row vector $\boldsymbol{W}_{\mathrm{o}}^{(i)}(j,:)$ which are multiplied to the exponentially weighted running averages of $k^{\mathrm{th}}$ recurrent statistic $\phi_{i,t}(k)$ corresponding to the $m$ timescales in $\mathcal{A}$ prior to being transformed by the neural unit generating the $j^{\mathrm{th}}$ predictive feature in $\mathbf{o}_{i,t}$. Compared to equation 4, the second penalty term in equation 5 promotes a group-sparse solution for $\boldsymbol{W}_{\mathrm{o}}^{(i)}$ to the effect that each predictive feature in $\mathbf{o}_{i,t}$ depends on only a few components of the recurrent statistic $\phi_{i,t}$ via their linearly mixed multi-scale exponentially weighted averages. We opine that the learned linear mixtures, represented by the intermediate products $\boldsymbol{W}_{\mathrm{o}}^{(i)}(j, \mathcal{G}_{j,k}) \mathbf{u}_{i,t}(\mathcal{G}_{k,j})$, are highly sensitive to certain past segments of the input time series $\mathbf{x}$. Consequently, the output features in $\mathbf{o}_{i,t}$ are both time-localized and component-specific, a common trait of real-world causal effects.

---

[1]Gaussian random matrices of appropriate dimensions are approximately isometries with overwhelming probability (Johnson & Lindenstrauss (1984)). However, instead of using $n$ independent instantiations of a Gaussian random matrix for initializing $\boldsymbol{D}_{\mathrm{r}}^{(i)}$, $1 \le i \le n$, we recommend initializing them with the same random matrix, as the latter strategy reduces the probability that any one of them is spurious encoder by $n$-fold.

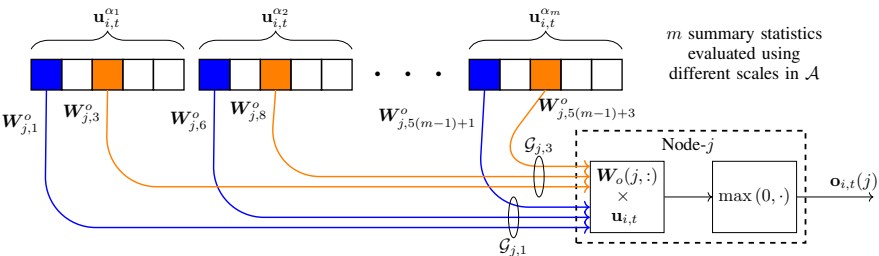

Figure 3: An illustration of the proposed group-wise mixing of the multi-timescale summary statistics $\mathbf{u}_{i,t}$ in the $i^{\text{th}}$ SRU (with $d_\phi = 5$) towards generating the $j^{\text{th}}$ predictive feature in $\mathbf{o}_{i,t}$. The weights corresponding to the same colored connections belong to the same group.

The above group-sparse regularization of the weight coefficients in $\boldsymbol{W}_{\text{o}}^{(i)}$, combined with the column-sparsity of $\boldsymbol{W}_{\text{in}}^{(i)}$, is pivotal to enforcing that the occurrence of any future pattern in a time series can be attributed to the past occurrences of a few highly time-localized patterns in the ancestral time series. The results of our numerical experiments further confirm that by choosing $\lambda_1$ and $\lambda_2$ appropriately, the proposed group-wise sparsity inducing regularization of $\boldsymbol{W}_{\text{o}}^{(i)}$ ameliorates overfitting, and the optimization in equation 5 yields an estimate of $\boldsymbol{W}_{\text{in}}^{(i)}$ whose column support closely reflects the true pairwise Granger causal relationships between the components of $\mathbf{x}$.

## 5 EXPERIMENTS

We evaluate the performance of the proposed SRU- and eSRU-based component-wise time series models in inferring pairwise Granger causal relationships in a multivariate time series. The proposed models are compared to the existing MLP- and LSTM-based models in Tank et al. (2018) and the attention-gated CNN-based model (referred hereafter as Temporal Causal Discovery Framework (TCDF)) in Nauta et al. (2019). To ensure parity between the competing models, the maximum size of all the input/hidden/output layers in the different NN/RNN time series models is fixed to 10, unless specified otherwise. The complete list of tuned hyperparameters of the considered models used for different datasets is provided in Appendix G. The performance of each method is qualified in terms of its AUROC (Area Under the Receiver Operating Characteristic curve). Here, the ROC curve illustrates the trade off between the true-positive rate (TPR) and the false-positive rate (FPR) achieved by the methods towards the detection of $n^2$ pairwise Granger causal relationships between the $n$ measured processes in the experiment. The ROC curves of SRU and eSRU models are obtained by sweeping through different values of the regularization parameter $\lambda_1$ in equation 4 and equation 5, respectively. Likewise, the ROCs of component-wise MLP and LSTM models are obtained by varying $\lambda_1$'s counterpart in Tank et al. (2018). For TCDF, the ROC curve is obtained by varying the threshold that is applied to attention scores of the trained AG-CNN model in Nauta et al. (2019).

### 5.1 LORENZ-96 SIMULATIONS

In the first set of experiments, the time series measurements $\mathbf{x}$ intended for Granger causal inference are generated according to the Lorenz-96 model which has been extensively used in climate science for modeling and prediction purposes (Schneider et al. (2017)). In the Lorenz-96 model of an $n$-variable system, the individual state trajectories of the $n$ variables are governed by the following set of odinary differential equations:

$$\frac{\partial \mathbf{x}_{t,i}}{\partial t} = -\mathbf{x}_{t,i-1}\left(\mathbf{x}_{t,i-2} - \mathbf{x}_{t,i+1}\right) - \mathbf{x}_{t,i} + F, \quad 1 \le i \le n. \tag{6}$$

where the first and the second terms on the RHS represent the *advection* and the *diffusion* in the system, respectively, and the third term $F$ is the magnitude of the external forcing. The system dynamics becomes increasingly chaotic for higher values of $F$ (Karimi & Paul (2010)). We evaluate and compare the accuracy of the proposed methods in inferring pairwise Granger causal relationships between $n = 10$ variables with Lorenz-96 dynamics. We consider two settings: $F = 10$ and $F = 40$ in order to simulate two different strengths of nonlinearity in the causal interactions between

Table 1: Averaged AUROC for $5$ independent Lorenz-96 datasets

(a) $F = 10$

| MODEL | AVERAGE AUROC | |
|---|---|---|
| | $T = 250$ | $T = 500$ |
| MLP | $0.93 \pm 0.02$ | $0.96 \pm 0.03$ |
| LSTM | $0.90 \pm 0.02$ | $0.95 \pm 0.05$ |
| TCDF | $0.70 \pm 0.01$ | $0.72 \pm 0.04$ |
| SRU | $0.84 \pm 0.03$ | $0.90 \pm 0.02$ |
| eSRU | $\mathbf{0.95 \pm 0.02}$ | $\mathbf{0.98 \pm 0.01}$ |

(b) $F = 40$

| MODEL | AVERAGE AUROC | |
|---|---|---|
| | $T = 250$ | $T = 500$ |
| MLP | $0.85 \pm 0.08$ | $0.94 \pm 0.03$ |
| LSTM | $0.78 \pm 0.09$ | $0.90 \pm 0.05$ |
| TCDF | $0.62 \pm 0.01$ | $0.68 \pm 0.04$ |
| SRU | $\mathbf{1.0 \pm 0.0}$ | $\mathbf{1.0 \pm 0.0}$ |
| eSRU | $0.99 \pm 0.0$ | $\mathbf{1.0 \pm 0.0}$ |

the variables. Here, the ground truth is straightforward i.e., for any $1 \leq i \leq n$, the $i^{\text{th}}$ component of time series $\mathbf{x}$ is Granger caused by its components with time indices from $i - 2$ to $i + 1$.

In the case of weak nonlinear interactions ($F = 10$), from Table 1a, we observe that eSRU achieves the highest AUROC among all competing models. The gap in performance is more pronounced when fewer time series measurements ($T = 250$) are available. In case of stronger nonlinear interactions ($F = 40$), we observe that both SRU and eSRU are the only models that are able to perfectly recover the true Granger causal network (Table 1b). Surprisingly, the SRU and eSRU models perform poorer when $F$ is small. This could be attributed to the proposed models not sufficiently regularized when fitted to weakly-interacting time series measurements that are less nonlinear.

## 5.2 VAR SIMULATIONS

In the second set of simulations, we consider the time series measurements $\mathbf{x}$ to be generated according to a $3^{\text{rd}}$ order linear VAR model:

$$\mathbf{x}_t = \boldsymbol{A}^{(1)}\mathbf{x}_{t-1} + \boldsymbol{A}^{(2)}\mathbf{x}_{t-2} + \boldsymbol{A}^{(3)}\mathbf{x}_{t-3} + \mathbf{w}_t, \quad t \geq 1, \tag{7}$$

where the matrices $\boldsymbol{A}^{(i)}, i = 1, 2, 3$ contain the regression coefficients which model the linear interactions between its $n = 10$ components. The noise term $\mathbf{w}_t$ is Gaussian distributed with zero mean and covariance $0.01\boldsymbol{I}$. We consider a sparse network of Granger causal interactions with only $30\%$ of the regression coefficients in $\boldsymbol{A}^i$ selected uniformly being non-zero and the regression matrices $\boldsymbol{A}^i$ being collectively joint sparse (same setup as in Bolstad et al. (2011)). All non-zero regression coefficients are set equal to $0.0994$ which guarantees the stability of the simulated VAR process.

From Table 2, we observe that all time series models generally achieve a higher AUROC as the number of measurements available increases. For $T = 500$, the component-wise MLP and the proposed eSRU are statistically tied when comparing their average AUROCs. For $T = 1000$, eSRU significantly outperforms the rest of the time series models and is able to recover the true Granger causal network almost perfectly.

Table 2: Averaged AUROC for $5$ independently generated VAR datasets

| MODEL | AVERAGE AUROC | |
|---|---|---|
| | $T = 500$ | $T = 1000$ |
| MLP | $\mathbf{0.94 \pm 0.03}$ | $0.93 \pm 0.02$ |
| LSTM | $0.79 \pm 0.12$ | $0.8 \pm 0.09$ |
| TCDF | $0.77 \pm 0.07$ | $0.78 \pm 0.04$ |
| SRU | $0.82 \pm 0.06$ | $0.91 \pm 0.04$ |
| eSRU | $0.93 \pm 0.05$ | $\mathbf{0.98 \pm 0.01}$ |

Table 3: Average AUROC corresponding to inferred brain connectivity for $5$ human subjects

| MODEL | AVERAGE AUROC |
|---|---|
| | $T = 200$ |
| MLP | $0.81 \pm 0.04$ |
| LSTM | $0.70 \pm 0.03$ |
| TCDF | $0.75 \pm 0.04$ |
| SRU | $0.78 \pm 0.02$ |
| eSRU | $\mathbf{0.84 \pm 0.03}$ |

## 5.3 IN SILICO ESTIMATION OF BRAIN CONNECTIVITY USING BOLD SIGNALS

In the third set of experiments, we apply the different learning methods to estimate the connections in the human brain from simulated *blood oxygenation level dependent* (BOLD) imaging data. Here, the individual components of $\mathbf{x}$ comprise $T = 200$ time-ordered samples of

the BOLD signals simulated for $n = 15$ different brain regions of interest (ROIs) in a human subject. To conduct the experiments, we use simulated BOLD time series measurements corresponding to the five different human subjects (labelled as 2 to 6) in the *Sim-3.mat* file shared at `https://www.fmrib.ox.ac.uk/datasets/netsim/index.html`. The generation of the *Sim3* dataset is described in Smith et al. (2011). The goal here is to detect the directed connectivity between different brain ROIs in the form of pairwise Granger causal relationships between the components of **x**.

From Table 3, it is evident that eSRU is more robust to overfitting compared to the standard SRU and detects the true Granger causal relationships more reliably. Interestingly, a single-layer cMLP model is found to outperform more complex cLSTM and attention gated-CNN (TCDF) models; however we expect the latter models to perform better when more time series measurements are available.

### 5.4 DREAM-3 IN SILICO NETWORK INFERENCE CHALLENGE

In the final set of experiments, we evaluate the performance of the different time series models in inferring gene regulation networks synthesized for the DREAM-3 In Silico Network Challenge (Prill et al. (2010); Marbach et al. (2009)). Here, the time series **x** represents the *in silico* measurements of the gene expression levels of $n = 100$ genes, available for estimating the gene regulatory networks of *E.coli* and *yeast*. A total of five gene regulation networks are to be inferred (two for E.coli and three for yeast) from the networks' gene expression level trajectories recorded while they recover from 46 different perturbations (each trajectory has 21 time points). All NN/RNN models are implemented with 10 neurons per layer, except for the componentwise MLP model which has 5 neurons per layer. From Table 4, we can observe that the proposed SRU and eSRU models are generally more accurate

Table 4: AUROCs for the inferred gene regulatory networks

| MODEL | AUROC | | | | |
|-------|---------|---------|---------|---------|---------|
| | E.coli-1 | E.coli-2 | Yeast-1 | Yeast-2 | Yeast-3 |
| MLP | 0.644 | 0.568 | 0.585 | 0.506 | 0.528 |
| LSTM | 0.629 | 0.609 | 0.579 | 0.519 | 0.555 |
| TCDF | 0.614 | 0.647 | 0.581 | 0.556 | **0.557** |
| SRU | 0.657 | **0.666** | 0.617 | **0.575** | 0.55 |
| eSRU | **0.66** | 0.629 | **0.627** | 0.557 | 0.55 |

compared to the MLP, LSTM, and attention-gated CNN (TCDF) models in inferring the true gene-gene interactions. For four out of the five gene regulatory networks, either SRU or eSRU was the best performing model among the competing ones.

## 6 CONCLUSION

In this work, we addressed the problem of inferring pairwise Granger causal relationships between stochastic processes that interact nonlinearly. We showed that the such causality between the processes can be robustly inferred from the regularized internal parameters of the proposed eSRU-based recurrent models trained to predict the time series measurements of the individual processes. Future work includes:

  i Investigating the use of other loss functions besides the mean-square error loss which can capture the exogenous and instantaneous causal effects in a more realistic way.

 ii Incorporating unobserved confounding variables/processes in recurrent models.

iii Inferring Granger causality from multi-rate time series measurements.

### ACKNOWLEDGEMENTS

This work is supported by a Singapore Ministry of Education (MOE) Tier 2 Grant (R-263-000-C83-112).

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

## A    REVIEW OF RELATED WORK

Initial efforts in testing for nonlinear Granger causality focused mostly on the nonparameteric approach. Baek & Brock (1992) proposed a general statistical test to detect nonlinear Granger causality between two variables under the assumption that the linear VAR modeling of their time series measurements results in i.i.d residual errors. The test involves computing correlation integral estimators of the conditional probabilities concerning distances between carefully selected lead-lag subsequences of the input bivariate time series. Successive works by Hiemstra & Jones (1994); Diks & Panchenko (2006); Bai et al. (2010); Diks & Wolski (2016) proposed their improved variants of

the Baek-Brock test. Hiemstra & Jones (1994) modified and extended the original Baek-Brock test to allow for weakly dependent residual errors in linear VAR modeling of the time series measurements. Diks & Panchenko (2006) proposed a rectified version of the Heimstra-Jones' test statistic, making it unbiased while fixing the issue of overrejection of the null hypothesis. Later works by Bai et al. (2010); Diks & Wolski (2016) extended the bi-variate test in Diks & Panchenko (2006) to the multivariate setting. The biggest common drawback of these nonparameteric tests is the large sample sizes required to robustly estimate the conditional probabilities that constitute the test statistic. Furthermore, the prevalent strategy in these methods of testing each one of the variable-pairs individually to detect pairwise Granger causality is unappealing from a computational standpoint, especially when a very large number of variables are involved.

In the model driven approach, the Granger causal relationships are inferred directly from the parameters of a data generative model fitted to the time series measurements. Compared to the nonparameteric approach, the model-based inference approach is considerably more sample efficient, however the scope of inferrable causal dependencies is dictated by the choice of data generative model. Nonlinear kernel based regression models have been found to be reasonably effective in testing of nonlinear Granger causality. Kernel methods rely on linearization of the causal interactions in a kernel-induced high dimensional feature space; the linearized interactions are subsequently modeled using a linear VAR model in the feature space. Based on this idea, Marinazzo et al. (2008) proposes a kernel Granger causality index to detect pairwise nonlinear Granger causality in the multivariate case. In Sindhwani et al. (2013); Lim et al. (2014), the nonlinear dependencies in the time series measurements are modeled using nonlinear functions expressible as sums of vector valued functions in the induced reproducing kernel Hilbert space (RKHS) of a matrix-valued kernel. In Lim et al. (2014), additional smoothness and structured sparsity constraints are imposed on the kernel parameters to promote consistency of the time series fitted nonlinear model. Shen et al. (2016) proposes a nonlinear kernel-based structural VAR model to capture instantaneous nonlinear interactions. The existing kernel based regression models are restrictive as they consider only additive linear combinations of the RKHS functions to approximate the nonlinear dependencies in the time series. Furthermore, deciding the optimal order of kernel based regression models is difficult as it requires prior knowledge of the mimimum time delay beyond which the causal influences are negligible.

By virtue of their universal approximation ability, RNNs offer a pragmatic way forward in modeling of complex nonlinear dependencies in the time series measurements for the purpose of inferring Granger causality. Mutiple recent works by Wang et al. (2018); Duggento et al. (2019); Abbasvandi & Nasrabadi (2019) have investigated the use of different types of RNNs for inferring nonlinear Granger causal relationships. However, they all adopt the same naïve strategy whereby each pairwise causal relationship is tested individually by estimating its causal connection strength. The strength of the causal connection from series $j$ to series $i$ is determined by the ratio of mean-squared prediction errors incurred by *unrestricted* and *restricted* RNN models towards predicting series $i$ using the past measurement sequences of all $n$ component including and excluding the $j^{\text{th}}$ component alone, respectively. The pairwise testing strategy however does not scale well computationally as the number of component series becomes very large. This strategy also fails to exploit the typical sparse connectivity of networked interactions between the processes which has unlocked significant performance gains in the existing linear methods (Bahadori & Liu (2013); Bolstad et al. (2011)).

In a recent work by Tank et al. (2018), the pairwise Granger causal relationships are inferred directly from the weight parameters of component-wise MLP or LSTM networks fitted to the time series measurements. By enforcing column-sparsity of the input-layer weight matrices in the fitted MLP/LSTM models, their proposed approach returns a sparsely connected estimate of the underlying Granger causal network. Due to its feedforward architecture, a traditional MLP network is not well-suited for modeling ordered data such as a time series. Tank et al. (2018) demonstrated that the MLP network can learn short range temporal dependencies spanning a few time delays by letting the network's input stage process multi-lag time series data over sliding windows. However, modeling long-range temporal dependencies using the same approach requires a larger sliding window size which entails an inconvenient increase in the number of trainable parameters. The simulation results in Tank et al. (2018) indicate that MLP models are generally outperformed by LSTM models in extracting the true topology of pairwise Granger causality, especially when the processes interact in a highly nonlinear and intricate manner. While purposefully designed for modeling short and long term temporal dependencies in a time series, the LSTM (Hochreiter & Schmidhuber (1997)) is very

general and often too much overparameterized and thus prone to overfitting. While using overparameterized models for inference is preferable when there is abundant training data available to leverage upon, there are several applications where the data available for causal inference is extremely scarce. It is our opinion that using a simpler RNN model combined with meaningful regularization of the model parameters is the best way forward in inferring Granger causal relationships from underdetermined time series measurements. Building on the ideas put forth by Tank et al. (2018), this paper investigates the use of Statistical Recurrent Units (SRUs) towards inferring Granger causality.

## B   PROXIMAL GRADIENT DESCENT UPDATES FOR ESTIMATING THE REGULARIZED WEIGHT PARAMETERS IN THE SRU AND $e$SRU MODELS

Noting that the proximal operator corresponding to mixed $\ell_1$-$\ell_2$ norm (group-norm) is the group-wise soft-thresholding operator, we use the following proximal gradient-descent updates to minimize the $i^{\text{th}}$ SRU's regularized loss in equation 4:

$$\boldsymbol{W}_{\text{in}}^{(i),t+1}(:,j) = S_{\lambda_1 \eta}\left(\boldsymbol{W}_{\text{in}}^{(i),t}(:,j) - \eta \nabla_{\boldsymbol{W}_{\text{in}}^{(i)}(:,j)} l_i(\Theta_{\text{SRU}}^{(i),t})\right), \ \ \forall j \in [n]. \tag{8}$$

Here, $l_i(\Theta_i^t) \triangleq \frac{1}{T-1}\sum_{t=1}^{T-1}(\hat{\text{x}}_{i,t} - \text{x}_{i,t+1})^2$ is the unregularized SRU loss function, $\eta$ is the gradient-descent stepsize and $S_{\lambda_1 \eta}$ is the elementwise soft-thresholding operator defined below.

$$S_{\lambda_1 \eta}(\boldsymbol{w}) \triangleq \begin{cases} \boldsymbol{w} - \lambda_1 \eta \frac{\boldsymbol{w}}{\|\boldsymbol{w}\|_2}, & \|\boldsymbol{w}\|_2 > \lambda_1 \eta \\ 0, & \|\boldsymbol{w}\|_2 \le \lambda_1 \eta \end{cases}, \ \ \forall \boldsymbol{w} \in \mathbb{R}^n. \tag{9}$$

The columns of weight matrix $\boldsymbol{W}_{\text{i}}^{(i)}$ in the $i^{\text{th}}$ eSRU model are also updated in exactly the same fashion as above.

Likewise, the $j^{\text{th}}$ row of the group-norm regularized weight matrix $\boldsymbol{W}_{\text{o}}^{(i)}$ in the eSRU optimization in equation 5 is updated as shown below.

$$\boldsymbol{W}_{\text{o}}^{(i),t+1}(j, \mathcal{G}_{j,k}) = S_{\lambda_2 \eta}\left(\boldsymbol{W}_{\text{o}}^{(i),t}(j, \mathcal{G}_{j,k}) - \eta \nabla_{\boldsymbol{W}_{\text{o}}^{(i)}(j, \mathcal{G}_{j,k})} l_i(\Theta_{\text{eSRU}}^{(i),t})\right), \ \ \forall j = 1, 2, \ldots d_{\text{o}}. \tag{10}$$

The gradient of the unregularized loss function $l_i, 1 \le i \le n$ associated with the SRU and eSRU models used in the above updates is evaluated via the backpropagation through time (BPTT) procedure (Jaeger (2002)).

## C   ABLATION STUDY

### C.1   CHOICE OF ENCODER FOR $e$SRU FEEDBACK'S STAGE-1: FIXED OR DATA DEPENDENT

As a possible further enhancement of the proposed eSRU time series model, one may consider learning the encoding map, $\boldsymbol{D}_r^{(i)}$, in the feedback path, as trainable parameters of the $i^{\text{th}}$ eSRU. In Table 5, we compare the Granger causality detection performance of this particular eSRU variant and the proposed design wherein $\boldsymbol{D}_r^{(i)}$ is taken to be a random matrix with i.i.d. Gaussian entries. The experimental setup is kept the same as in Section 5, and the entries of $\boldsymbol{D}_r^{(i)}$ in the eSRU variant are $\ell_2$-norm penalized during training.

We observe that the performance of these two models is statistically tied, which indicates that the randomly constructed $\boldsymbol{D}_{\text{r}}^{(i)}$ is able to distill the necessary information from the high-dimensional summary statistics $\boldsymbol{u}_{i,t-1}$ required for generating the feedback $\boldsymbol{r}_{i,t}$. Based on these results, we recommend using the proposed eSRU design with its randomly constructed encoding map $\boldsymbol{D}_{\text{r}}^{(i)}$, because of its simpler design and reduced training complexity.

### C.2   IMPACT OF GROUP-SPARSE REGULARIZATION OF $\boldsymbol{W}_{\text{o}}^{(i)}$

In order to highlight the importance of learning time-localized predictive features in detecting Granger causality, we compare the following two time series models:

Table 5: Average AUROC for eSRU variants

| DATASET | Average AUROC | |
| --- | --- | --- |
| | Randomly constructed $D_r^{(i)}$ | $D_r^{(i)}$ as trainable parameters |
| Lorenz ($T = 250, F = 10$) | $0.95 \pm 0.02$ | $0.97 \pm 0.01$ |
| Lorenz ($T = 500, F = 10$) | $0.98 \pm 0.01$ | $0.99 \pm 0.0$ |
| Lorenz ($T = 250, F = 40$) | $0.99 \pm 0.0$ | $0.98 \pm 0.01$ |
| Lorenz ($T = 500, F = 40$) | $1.0 \pm 0$ | $1.0 \pm 0.0$ |
| VAR ($T = 500$) | $0.93 \pm 0.05$ | $0.91 \pm 0.04$ |
| VAR ($T = 1000$) | $0.98 \pm 0.01$ | $0.98 \pm 0.01$ |
| NetSim | $0.84 \pm 0.03$ | $0.80 \pm 0.02$ |

i. proposed eSRU (with group-sparse regularization of $W_o^{(i)}$ as described in Section 4.2)

ii. eSRU variant with ridge-regularized $W_o^{(i)}$

Once again, we use the same experimental settings as mentioned in Section 5. From Table 6, we observe that barring the `Lorenz-96` ($T$=250/500, $F$=40) datasets, for which nearly perfect recovery of the Granger causal network is achieved, the average AUROC improves consistently for the other datasets by switching from unstructured ridge regularization to the proposed group-sparse regularization of the output weight matrix $W_o^{(i)}$.

Table 6: Performance of $e$SRU variants with different regularizations of $W_o^{(i)}$

| DATASET | Average AUROC | |
| --- | --- | --- |
| | Proposed group-sparse regularization of $W_o^{(i)}$ | Ridge regularization of $W_o^{(i)}$ |
| Lorenz ($T = 250, F = 10$) | $0.95 \pm 0.02$ | $0.93 \pm 0.03$ |
| Lorenz ($T = 500, F = 10$) | $0.98 \pm 0.01$ | $0.94 \pm 0.04$ |
| Lorenz ($T = 250, F = 40$) | $0.99 \pm 0.0$ | $0.99 \pm 0.0$ |
| Lorenz ($T = 500, F = 40$) | $1.0 \pm 0.0$ | $1.0 \pm 0.0$ |
| VAR ($T = 500$) | $0.93 \pm 0.05$ | $0.90 \pm 0.03$ |
| VAR ($T = 1000$) | $0.98 \pm 0.01$ | $0.96 \pm 0.01$ |
| NetSim | $0.84 \pm 0.03$ | $0.83 \pm 0.03$ |

## D  IMPLEMENTATION DETAILS

- **Activation function for SRU and eSRU models**
  While the standard SRU proposed by Oliva et al. (2017) uses ReLU neurons, we found in our numerical experiments that using the Exponential Linear Unit (ELU) activation resulted in better performance. The ELU activation function is defined as

$$ELU(x) = \begin{cases} x & x > 0 \\ \alpha(e^x - 1) & x \leq 0 \end{cases}, \quad \alpha > 0. \tag{11}$$

  In our simulations, the constant $\alpha$ is set equal to one.

- **Number of neural layers in SRU model**
  To approximate the generative functions $f_i$ in equation 1, we consider the simplest architecture for the SRU networks, whereby the constituent ReLU networks generating the recurrent features, output features and feedback have a single layer feedforward design with equal number of neurons.

- **Number of neural layers in Economy-SRU model**
  The ReLU networks used for generating the recurrent and output features in the proposed

eSRU model have a single-layer feedforward design. However, the second stage of eSRU's modified feedback can be either single or multi-layered feedforward network. Provided that $d'_r \ll md_\phi$, a multi-layer implementation of the second stage of eSRU's feedback can still have fewer trainable parameters overall compared to the SRU's single layer feedback network. The simulation results in Section 5 are obtained using a two-layer ReLU network in the second stage of eSRU's feedback for the DREAM-3 experiments, and while using a three-layer design for the Lorenz-96, VAR and NetSim experiments.

- **Self-interactions in Dream-$3$ gene networks**
  The in-silico gene networks synthesized for the DREAM-3 challenge have no self-connections. Noting that none of the Granger causal inference methods evaluated in our experiments intentionally suppress the self-interactions, the reported AUROC values are computed by ignoring any self-connections in the inferred Granger causal networks.

## E   PYTHON CODES

**cMLP & cLSTM models**
Pytorch implementation of the componentwise MLP and LSTM models are taken from `https://github.com/icc2115/Neural-GC`.

**Temporal Causal Discovery Framework (TCDF)**
Pytorch implementation of the attention-gated CNN based Temporal Causal Discovery Framework (TCDF) is taken from `https://github.com/M-Nauta/TCDF`.

**Proposed SRU and Economy-SRU models**
Pytorch implementations of the proposed componentwise SRU and eSRU models are shared at `https://github.com/sakhanna/SRU_for_GCI`.

## F   ROC PLOTS

The receiver operating characteristics (ROC) of different Granger causal inference methods are compared in Figures 4-7. Here, an ROC curve represents the trade-off between the true-positive rate (TPR) and the false-positive rate (FPR) achieved by a given method while inferring the underlying pairwise Granger causal relationships.

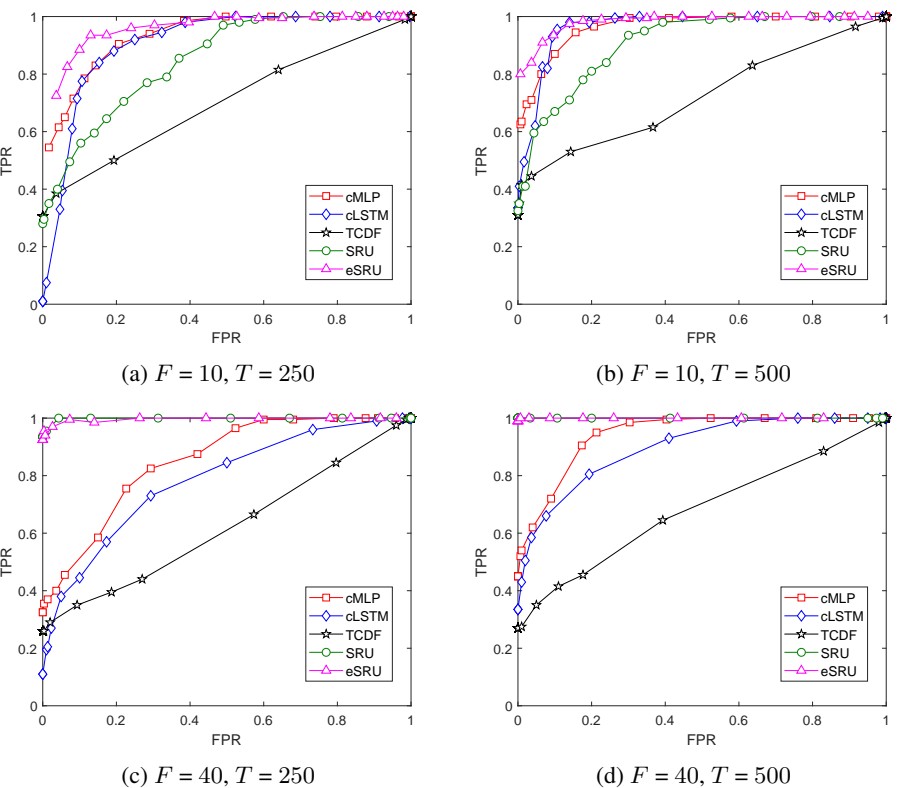

(a) $F = 10$, $T = 250$        (b) $F = 10$, $T = 500$

(c) $F = 40$, $T = 250$        (d) $F = 40$, $T = 500$

Figure 4: Average ROC curves for Lorenz-96 datasets

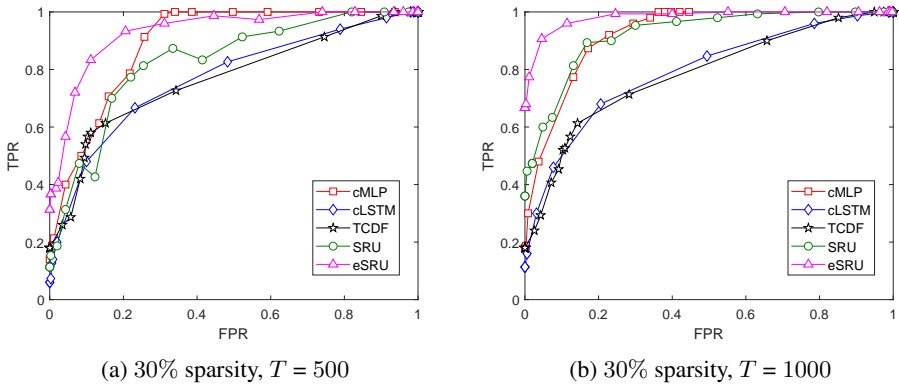

(a) 30% sparsity, $T = 500$        (b) 30% sparsity, $T = 1000$

Figure 5: Average ROC curves for VAR datasets

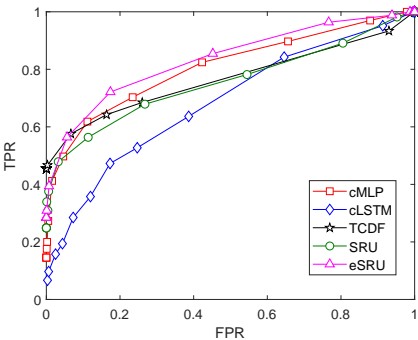

Figure 6: Average ROC curves for the NetSim experiment.

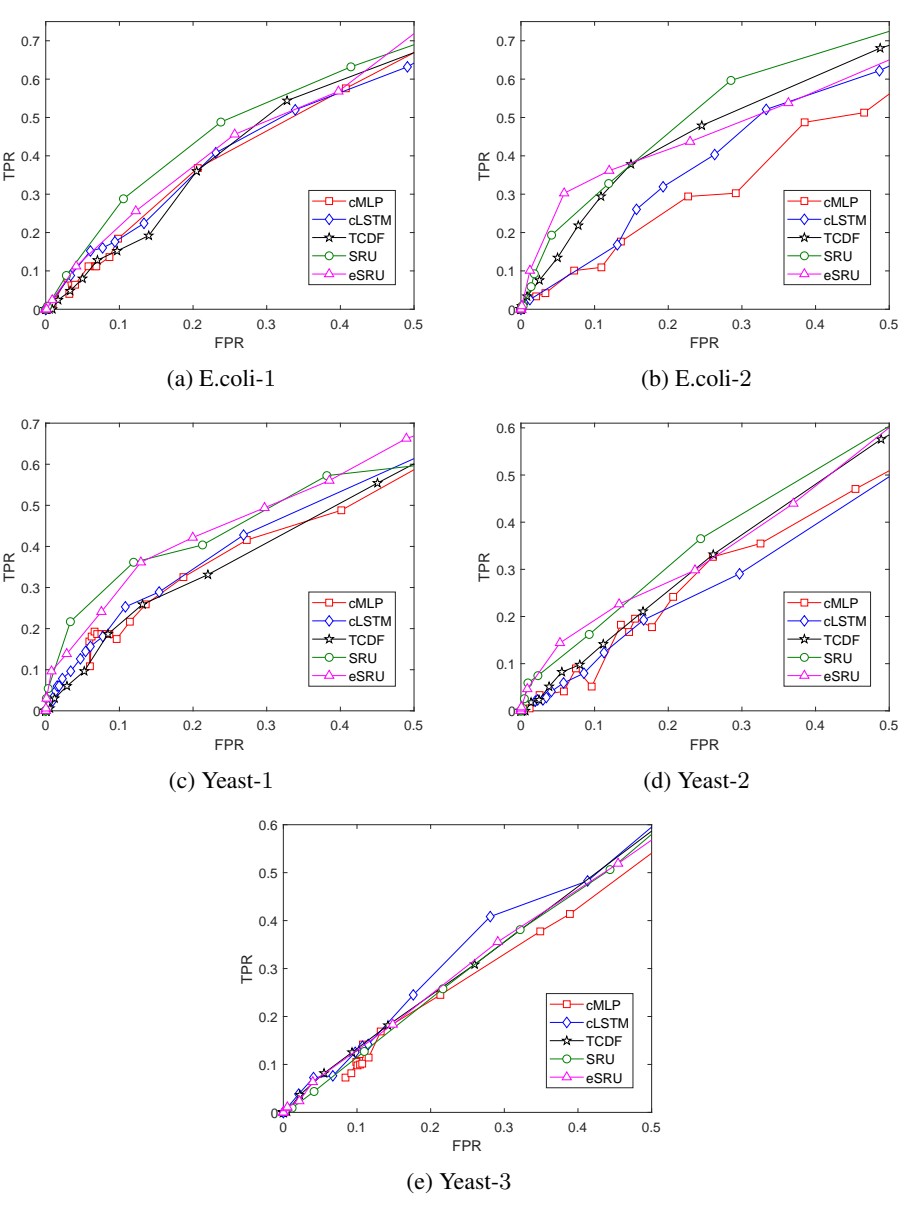

(a) E.coli-1

(b) E.coli-2

(c) Yeast-1

(d) Yeast-2

(e) Yeast-3

Figure 7: ROC curves for Dream-3 datasets

## G    TUNED HYPERPARAMETERS

Tables 7 to 11 summarize the chosen hyperparameters and configurations of the different NN/RNN models used for generating the results reported in Section 5. For the Dream-3 experiments, the model configurations used for inferring the *E.coli*-1 gene regulatory network Prill et al. (2010) have been provided in the tables.

| Parameters | Tuning range | Dataset | | | |
|---|---|---|---|---|---|
| | | Lorenz (F = 10/40) | VAR | Dream-3 | NetSim |
| # Neural units per layer | − | 10 | 10 | 5 | 10 |
| Batch size | − | 250/500/1000 | 500/1000 | 21 | 200 |
| Learning rate | Two-fold cross-validation across [5e-5, 1e-1] | 0.0005 ($F = 10$), 0.001 ($F = 40$) | 0.1 | 0.0005 | 0.0005 |
| Ridge regularization bias | Two-fold cross-validation across [5e-5, 10] | 0.232079 ($F = 10$), 10.0 ($F = 40$) | 0.002 | 5.0 | 0.464159 |
| Block-sparse regularization bias (input layer) | − | $[0.1, 10]$ ($F = 10$), $[1, 100]$ ($F = 40$) | $[0.0001, 0.01]$ | $[0.1, 100]$ | $[0.1, 3.162]$ |
| # Lags | − | 5 | 5 | 2 | 5 |
| # Training epochs | − | 2000 | 2000 | 2000 | 2000 |

Table 7: Componentwise MLP model configuration (Refer Tank et al. (2017) for detailed description of the model parameters).

| Parameters | Tuning strategy | Dataset | | | |
|---|---|---|---|---|---|
| | | Lorenz (F = 10/40) | VAR | Dream-3 | NetSim |
| # units per layer | − | 10 | 10 | 10 | 10 |
| Batch size | − | 250/500/1000 | 500/1000 | 21 | 200 |
| Learning rate | Two-fold cross-validation across [5e-5, 1e-1] | 0.0005 ($F = 10$), 0.001 ($F = 40$) | 0.1 | 0.0005 | 0.001 |
| Ridge regularization bias | Two-fold cross-validation across [5e-5, 10] | 0.021544 ($F = 10$), 5.0 ($F = 40$) | 0.0005 | 5.0 | 0.010772 |
| Group-sparse regularization bias | − | $[1, 56.234]$ | $[0.003162, 0.01]$ | $[0.1, 17.52]$ | $[0.1, 3.162]$ |
| Truncation | − | No | No | No | No |
| # Training epochs | − | 2000 | 2000 | 2000 | 4000 |

Table 8: Componentwise LSTM model configuration (Refer Tank et al. (2017)) for detailed description of the model parameters).

| Parameters | Tuning strategy | Dataset | | | | |
|---|---|---|---|---|---|---|
| | | Lorenz (F = 10) | Lorenz (F = 40) | VAR | Dream-3 | NetSim |
| Kernel size | $\{2, 4\}$ | 4 | 2 | 2 | 2 | 4 |
| Batch size | − | 250/500/1000 | 250/500/1000 | 500/1000 | 21 | 200 |
| Layers | $\{2, 3, 4\}$ | 2 | 2 | 3 | 3 | 2 |
| Learning rate | $\{10^{-1}, 10^{-2}, 10^{-3}\}$ | 0.01 | 0.01 | 0.001 | 0.01 | 0.001 |
| Dilation | $\{1, 2, 4\}$ | 1 | 2 | 1 | 4 | 2 |
| Significance | − | 0.8 | 0.8 | 8 | 0.8 | 0.8 |
| # Training epochs | $\{1000, 2000, 4000\}$ | 2000 | 2000 | 2000 | 4000 | 2000 |

Table 9: TCDF's Attention-gated CNN model parameters (Refer Nauta et al. (2019) for detailed description of the model parameters).

| Parameters | Tuning strategy | Dataset | | | |
|---|---|---|---|---|---|
| | | Lorenz (F = 10/40) | VAR | Dream-3 | NetSim |
| # units per layer | — | 10 | 10 | 10 | 10 |
| $\mathcal{A}$ | — | $\{0.0, 0.01, 0.1, 0.99\}$ | $\{0.0, 0.01, 0.1, 0.99\}$ | $\{0.0, 0.01, 0.1, 0.5, 0.99\}$ | $\{0.0, 0.01, 0.1, 0.99\}$ |
| Learning rate | Two-fold cross-validation across $[5e\text{-}4, 1e\text{-}1]$ | $0.005\ (F = 10)$, $0.01\ (F = 40)$ | 0.04 | 0.005 | 0.001 |
| Ridge regularization bias | Two-fold cross-validation across $[0.01, 10]$ | $0.021544\ (F = 10)$, $0.464159\ (F = 40)$ | 0.021544 | 0.2 | 0.464159 |
| Group-sparse regularization bias $\lambda_1$ (input layer) | — | $[0.1, 1]\ (F = 10)$, $[0.0631, 1]\ (F = 40)$ | $[0.001, 1]$ | $[0.01, 1.0]$ | $[0.1, 3.162]$ |
| Batch size | — | 125 | 125 | 21 | 5 |
| # Training epochs | — | 2000 | 2000 | 1000 | 2000 |

Table 10: Componentwise SRU model configuration

| Parameters | Tuning strategy | Dataset | | | |
|---|---|---|---|---|---|
| | | Lorenz (F = 10/40) | VAR | Dream-3 | NetSim |
| # units per layer | — | 10 | 10 | 10 | 10 |
| $\mathcal{A}$ | — | $\{0.0, 0.01, 0.1, 0.99\}$ | $\{0.0, 0.01, 0.1, 0.99\}$ | $\{0.05, 0.1, 0.2, 0.99\}$ | $\{0.0, 0.01, 0.1, 0.99\}$ |
| Learning rate | Two-fold cross-validation across $[5e\text{-}4, 1e\text{-}1]$ | 0.01 | 0.01 | 0.001 | 0.001 |
| Ridge regularization parameter | Two-fold cross-validation across $[0.01, 10]$ | $0.001\ (F = 10)$, $0.043088\ (F = 40)$ | 0.021544 | 0.1 | 0.232 |
| Group-sparse regularization bias $\lambda_1$ (input layer) | — | $[0.03162, 0.1]$ | $[0.03162, 0.3162]$ | $[0.1, 3.162]$ | $[0.1, 3.162]$ |
| Group-sparse regularization bias $\lambda_2$ (output feature layer) | Two-fold cross-validation across $[0.01, 10]$ | $0.232079\ (F = 10)$, $0.928318\ (F = 40)$ | 0.464159 | 1.0 | 0.005 |
| # layers in the second stage of feedback network | — | 2 | 2 | 1 | 2 |
| Batch size | — | 125 | 125 | 21 | 5 |
| # Training epochs | — | 2000 | 2000 | 2000 | 2000 |

Table 11: Economy-SRU model configuration

