# OpenReview forum: "Economy Statistical Recurrent Units For Inferring Nonlinear Granger Causality"
_ICLR.cc/2020/Conference — Accept (Poster)_

### Official Review · AnonReviewer3 · 2019-10-23
**Official Blind Review #3**

**Rating:** 6

**Review:**

The paper proposed to use SRU for inferring nonlinear granger causality. It also provided two extensions of SRU with regularization to alleviate the issue of overfitting.

SRU was proposed in a previous paper. This paper extended this algorithm for inferring granger causality through applying group sparse regularization, which is a pretty smart design to me. Another major innovation comes from the two modifications to combat the possible overfitting when using very fine-grained scale parameters in SRU.

The paper was well-written in general. I can follow the paper without problem.
The effectiveness of the algorithm depends on the sparse regularization, which lacks theoretical guarantee for non-convex optimization. Based on the experimental results, it doesn't seem a big problem though. But I'd like to see some analysis of the actual sparsity of the weight with the proposed group sparse regularization.

The experiments seem convincing to me and I really appreciate the authors provided the tuned hyper-parameters in the appendix.


**Experience Assessment:**

I have read many papers in this area.

**Review Assessment: Checking Correctness Of Derivations And Theory:**

I assessed the sensibility of the derivations and theory.

**Review Assessment: Checking Correctness Of Experiments:**

I assessed the sensibility of the experiments.

**Review Assessment: Thoroughness In Paper Reading:**

I read the paper at least twice and used my best judgement in assessing the paper.

---

> ### Author Response · Authors · 2019-11-14
> **Response to reviewer #3**
>
> We thank the reviewer for the useful comments. Please find our pointwise responses below.
>
> >> The effectiveness of the algorithm depends on the sparse regularization, which lacks theoretical guarantee for non-convex optimization. Based on the experimental results, it doesn't seem a big problem though. But I'd like to see some analysis of the actual sparsity of the weight with the proposed group sparse regularization.
>
> [Reply]
> As suggested by the reviewer, to gain a better understanding of the impact of the group-sparse penalization of $W_{o}$ , we have conducted an ablation study comparing eSRU’s performance in the following two cases:
> (i)	proposed group-sparse penalization of $W_{o}$
> (ii)	unstructured ridge regularization of $W_{o}$.
> The average AUROC in detecting Granger causality achieved by these two implementations of the eSRU model are reported in the following table.
>
> --------------------------------------------------------------------------------------------------------
>                                               | Proposed group sparse       | Ridge regularization
> Dataset                                | regularization of $W_{o}$   | of $W_{o}$
> --------------------------------------------------------------------------------------------------------
> Lorenz (T = 250, F = 10)    |   0.95 ± 0.02                            |   0.93 ± 0.03
> Lorenz (T = 500, F = 10)    |   0.98 ± 0.01                            |   0.93 ± 0.04
> Lorenz (T = 250, F = 40)    |   0.99   ± 0.0                            |   0.99 ± 0.0
> Lorenz (T = 500, F = 40)    |   1.0 ± 0.0                                |   1.0 ± 0.0
> VAR (T = 500)                      |   0.93 ± 0.05                           |   0.90 ± 0.03
> NetSim                                |   0.84 ± 0.03                           |   0.83 ± 0.03
> -------------------------------------------------------------------------------------------------------
>
> In the above results, we can see that the average AUROC improves in all cases (barring the Lorenz, T = 250/500, F=40 dataset for which there is near perfect recovery of Granger causal network) by switching from unstructured ridge regularization to the proposed group-sparse regularization of the output weight matrix $W_{o}$.
>
> Performing a theoretical analysis of the group-sparse regularization for this model is out of the scope of the paper. The above ablation study is hopefully sufficiently convincing in reinforcing the idea that the proposed group-sparsity in the output weight matrix generally improves the performance.
>
> The revised manuscript discusses the above ablation study in the newly added Appendix C.2.

---

### Official Review · AnonReviewer1 · 2019-10-24
**Official Blind Review #1**

**Rating:** 6

**Review:**

In this paper the authors propose using Statistical Recurrent Units to predict the network for Granger causality. They motivate this choice by the high representation power of SRUs for multivariate time series, the good performance they usually enjoy and as a way to alleviate the vanishing gradient problem. More importantly the particular form of the SRUs gives a very simple predictor and therefore explanability for Granger causality: the authors propose to simply mark serie $i$ as Granger caused by $j$ if the $j$th column of the input mixing matrix of the $g_i$ is non-zero.
In order to force whole columns of the input matrix to be negative, the authors use a group regularization on the columns of the input matrix $W_{\text{in}}$. The resulting problem is then optimized by proximal gradient descent.
The second contribution of the authors is eSRU, a smaller (in terms of parameters) variant of SRU in order to prevent overfitting.
First the authors introduce a dimensionality reduction layer before the SRU units by using fixed non-trainable random projections. Then the authors propose adding a regularization term for the output mixing matrix, which represents the bulks of the parameters of the model.
The eSRU and SRU causal models presented are then compared to the previous state of the art on several datasets, both synthetic and real, where they manage to reach a new state of the art.
Regarding the two improvements in eSRU proposed by the authors, I have several questions:
First, regarding the random projections, have the authors tried learning the projections ? While it increases the risk of overfitting it is possible, with enough regularization, that it may help learn good representations. Secondly, as I understand it a single projection is drawn at random for every component. While we know from Johnson-Lindenstrauss's Lemma that this will in average be a good strategy, how stable is the model to spurious projections ?
Regarding the scarification of the output matrix: the number of parameters is effectively reduced but the computational requirements are still the same. Did the authors try any explicit methods, maybe matrix factorization, to exploit the very specific sparse structure of $W_o$ and reduce the number of operations and parameters ?
Finally, while the results are undeniable, just observing the non-zero columns of $W_\text{in}$ is as I understand it a mostly ad hoc rule. It would have been interesting to empirically verify its validity by contrasting it with the results given by following equation 2.
Overall the paper presents an interesting model for inferring Granger causality. The authors clearly present their two main contributions and verify them using varied datasets.
The authors also made a clear and appreciated effort toward reproducibility by including all hyperparameters and implementation details as well as the code, including those of competing techniques.
The inclusion of the literature review in the appendix is also greatly appreciated.

**Experience Assessment:**

I have read many papers in this area.

**Review Assessment: Checking Correctness Of Derivations And Theory:**

I carefully checked the derivations and theory.

**Review Assessment: Checking Correctness Of Experiments:**

I carefully checked the experiments.

**Review Assessment: Thoroughness In Paper Reading:**

I read the paper thoroughly.

---

> ### Author Response · Authors · 2019-11-14
> **Response to reviewer #1 (Part-1)**
>
>  We thank the reviewer for the detailed comments and suggestions. Please find our point by point responses below.
>
> >> First, regarding the random projections, have the authors tried learning the projections? While it increases the risk of overfitting it is possible, with enough regularization, that it may help learn good representations.
>
> [Reply]
> It is difficult to surmise up front if learning the encoder $D_{r}$ as trainable parameters will lead to better performance compared to the proposed approach wherein $D_{r}$ is taken to be a fixed Gaussian map.  To gain clarity on this, we conducted an ablation study to gauge the relative performance of these two implementations of the eSRU model. The following table reports the average AUROC obtained for different datasets.
>
>  Dataset:                                         Randomly constructed                 Encoder Dr as trainable
>                                                           encoder Dr                                       parameters
> ========================================================================
> Lorenz (T = 250, F = 10)	              0.95 ± 0.02                                      0.97 ± 0.01
> Lorenz (T = 500, F = 10) 	              0.98 ± 0.01                                      0.99 ± 0.0
> Lorenz (T = 250, F = 40) 	              0.99 ± 0.0                                        0.98 ± 0.01
> Lorenz (T = 500, F = 40) 	              1.0 ± 0                                             1.0 ± 0.0
> VAR (T = 500)                                      0.93 ± 0.05                                      0.91 ± 0.04
> VAR (T = 1000)                                    0.98 ± 0.01                                      0.98 ± 0.01
> NetSim                                                0.84 ± 0.03                                      0.80 ± 0.02
>
> We find that the two approaches are statistically tied in performance, which suggests that the random projections are able to successfully distill the necessary information from the high-dimensional sufficient statistics required for generating the feedback. Therefore, our recommendation for the final eSRU design is to avoid learning the encoder matrix $D_{r}$ as trainable parameters and instead use a randomly initialized $D_{r}$ to reduce the overall training complexity.
>
> In the revised manuscript, we have added a new appendix (C.1) which discusses the above ablation study in more detail.
>
>
> >> Secondly, as I understand it a single projection is drawn at random for every component. While we know from Johnson-Lindenstrauss's Lemma that this will in average be a good strategy, how stable is the model to spurious projections?
>
> [Reply]
> Good point! We agree with the reviewer that the strategy of drawing $D_{r}$ from a Gaussian ensemble may occasionally result in spurious instantiations which are characteristically bad JL transforms. However, the probability that a randomly drawn $D_{r}$ is a bad JL transform can be made arbitrary small by choosing a sufficiently high value for the dimension $d_{r}^{\prime}$.
>
> Nonetheless, the issue can be mitigated significantly by using the *same* encoder $D_{r}$ for each of the n SRUs predicting the distinct time series components. By doing so, the probability that $D_{r}$ (now drawn only once from a Gaussian ensemble) turns out to be a bad JL transform reduces by n-fold, where n is the number of time series.
>
> The revised manuscript now includes a footnote on page-6 highlighting this issue and recommending the use of a common encoder $D_{r}$ across the n SRUs.

---

> ### Author Response · Authors · 2019-11-14
> **Response to reviewer #1 (Part-2)**
>
> Continuing from part-1...
>
> >> Regarding the scarification of the output matrix: the number of parameters is effectively reduced but the computational requirements are still the same. Did the authors try any explicit methods, maybe matrix factorization, to exploit the very specific sparse structure of $W_{o}$ and reduce the number of operations and parameters?
>
> [Reply]
> We thank the reviewer for this interesting suggestion. Splitting the weight matrix $W_{o}$ into low-rank factor matrices will certainly reduce the number of trainable parameters, however, the goal here is not just limited to reducing the number of trainable parameters but also to promote a specific mixing of the multi-scale summary statistics in order to learn time localized predictive features. While, one can certainly serve the latter goal by group-sparse regularizing the coefficients in the right-sided factor matrix ($Z$ in $W_{o} = H Z$), there is an inadvertent loss in expressibility of the predictive features in the feature vector $o_{t}$ which is now restricted to lie in an $r$-dimensional subspace ($r$ is the rank of the factor matrices H and Z, which by design is typically much smaller than $d_{o}$). Furthermore, the unnecessary coupling/mixing due to the dense factor matrix $H$ results in generation of correlated features in $o_{t}$ which will be difficult to interpret.
>
> In our opinion, the proposed strategy based on the use of a mixed $\ell_{1}-\ell_{2}$ norm penalty is the most straightforward way to implement the desired mixing of the summary statistics’ components in order to promote learning of time-localized predictive feature/component in $o_{t}$. The proposed regularization strategy also preserves the interpretability of the predictive features in the sense that for the $i^{\text{th}}$ predictive feature, its sensitivity to past measurements can be profiled across time by analyzing the weights in the active groups in the $i^{\text{th}}$ row of $W_{o}$.
>
>
>
> >> Finally, while the results are undeniable, just observing the non-zero columns of Win is as I understand it a mostly ad hoc rule. It would have been interesting to empirically verify its validity by contrasting it with the results given by following equation 2
>
> [Reply]
> We would like to clarify that the strategy of deducing the Granger causal components from the sparse column support of the input weight matrices $W_{in}^{(i)}$ in the individual eSRU models is not ad hoc but  in fact directly motivated from the condition in equation-2 as explained below.
>
> Under the assumption that the true Granger causal network is sparsely connected, the function invariance condition in equation-2 translates to input-selectivity of the unknown generative function $f_{i}$ , i.e., $f_{i}$’s output depends on the input past measurements of only a small subset of the total n component series. This viewpoint guides us to learn an economy-SRU model which closely approximates $f_{i}$’s output, while simultaneously being input-selective. From equation-3b, it is easy to see that the column support of the input weight matrix $W_{in}^{(i)}$ dictates exactly which subset of the n components series ultimately play a role in the generation of the recurrent statistics and the predicted future samples of the $i^{\text{th}}$ series. In other words, column sparsity of $W_{in}^{(i)}$  is equivalent to input-selectivity of the $i^{\text{th}}$ SRU model.

---

### Official Review · AnonReviewer2 · 2019-10-24
**Official Blind Review #2**

**Rating:** 6

**Review:**

the paper attempts to infer Granger causality between nonlinearly interacting stochastic processes from their time series measurements. instead of using MLP/LSTM etc to to model time series measurement, the paper proposed to use component-wise time series prediction model with Statistical Recurrent Units to model the measurements. they consider a low-dimensional version of SRU, which they call economy-SRU. in particular, they use group-wise regularizing to accompany the particular structure of the model to aid interpretability. they compared the performance with existing models with MLP/LSTM and show some gains in a few examples (but not all.) the proposal is interesting, but the experiment section might need further strengthening. currently, the experimental results do not immediately pop out as showing eSRU particularly useful.

**Experience Assessment:**

I do not know much about this area.

**Review Assessment: Checking Correctness Of Derivations And Theory:**

I assessed the sensibility of the derivations and theory.

**Review Assessment: Checking Correctness Of Experiments:**

I assessed the sensibility of the experiments.

**Review Assessment: Thoroughness In Paper Reading:**

I read the paper at least twice and used my best judgement in assessing the paper.

---

> ### Author Response · Authors · 2019-11-14
> **Response to reviewer #2**
>
> We thank the reviewer for the useful feedback. Please find our pointwise responses below.
>
> >> They compared the performance with existing models with MLP/LSTM and show some gains in a few examples (but not all.)
>
> [Reply]
> We would like to emphasize here that the proposed economy-SRU model is the top performing model for 9 out of the 12 datasets, and among the top-2 models for 11 out of the 12 datasets considered in our experiments. The tables below provide a detailed breakdown of the top performing models.
>
> Model:                                            In top-1     In top-2
> =============================================
> MLP                                                   1/12         9/12
> LSTM                                                 0/12         1/12
> Attention Gated-CNN (TCDF)       2/12         3/12
> (Proposed) Economy-SRU            9/12         11/12
>
> (*As the SRU model is only an intermediate step in the development of the proposed eSRU model, we exclude it from our comparisons)
>
> List of best and second-best performing models for different datasets
> Dataset                                  Best model (Avg. AUROC)             Second-best model (Avg. AUROC)
> ==================================================================================
> Lorenz (F = 10, T = 250)        Economy-SRU (0.95)                        MLP (0.93)
> Lorenz (F = 10, T = 500)        Economy-SRU (0.98)                        MLP (0.96)
> Lorenz (F = 40, T = 250)        Economy-SRU (0.99)                        MLP (0.85)
> Lorenz (F = 40, T = 500)        Economy-SRU (1.0)                          MLP (0.94)
> VAR (T = 500)                         MLP (0.94)                                          Economy-SRU (0.93)
> VAR (T = 1000)                       Economy-SRU (0.98)                         MLP (0.93)
> Netsim (BOLD-FMRI)           Economy-SRU (0.84)                         MLP (0.81)
> Dream-3 (Ecoli-1)                 Economy-SRU (0.66)                         MLP (0.644)
> Dream-3 (Ecoli-2)                 AG-CNN (0.647)                                 Economy-SRU (0.629)
> Dream-3 (Yeast-1)                Economy-SRU (0.627)                      MLP (0.585)
> Dream-3 (Yeast-2)                Economy-SRU (0.557)                      AG-CNN (0.556)
> Dream-3 (Yeast-3)                AG-CNN (0.557)                                 LSTM (0.555)               ( eSRU (0.55) in third place)
>
> Based on the above performance breakdown, it is fair to conclude that the proposed economy-SRU model with its unique ability to learn time-localized predictive features comprehensively outperforms the MLP, LSTM and AG-CNN models in detecting nonlinear Granger causality. It should be further noted that for the three datasets where the proposed eSRU model is not the best, the relative drop in the avg. AUROC against the top performing model is marginal.
>
>
>
> >> the proposal is interesting, but the experiment section might need further strengthening. currently, the experimental results do not immediately pop out as showing eSRU particularly useful.
>
> [Reply]
> While the performance gains from using the economy-SRU model may not be exceptionally better across the board, there is > 5-16 % improvement in avg. AUROC observed for Lorenz (F40, T 250/500), VAR (T=1000) and Dream-3 (Yeast-1) datasets.
>
> Nonetheless, the key idea that we would like to stress in this paper is that most time series encountered in practice have the distinguishing feature that the occurrence of any future pattern in a time series can be attributed to the past occurrences of a few time-localized patterns in the ancestral time series. The proposed eSRU model is designed specifically to enforce such a causal prescription. in order to learn time-localized predictive features for the purpose of predicting the future time series samples. Our experiments reinforce the veracity of the above causal description, especially since the eSRU model is found to generally outperform the existing MLP/LSTM/Attention-Gated CNN models (which do not explicitly enforce time-localization of the learned predictive features).
> In page-6 of the revised manuscript, we have added a sentence to emphasize the above causal rule as the motivation behind one of the SRU modifications.

---

### Author Response · Authors · 2019-11-15
**Summary of revision**

We have uploaded a revision of the paper with the following updates:

1. Added appendix (C.1) containing new experiments to compare different design choices for the encoder $D_{r}$ in the eSRU design, as part of our response to a comment by reviewer #1.

2.  Added a sentence (in page-6, last line) to motivate one of the SRU modifications as part of our response to a comment by reviewer #2.

3. Added an ablation study (as appendix (C.2)) to demonstrate the impact of group-sparse regularization of the weight matrix $W_{o}$ on eSRU's performance, as advised by reviewer #3.

4. Added a footnote in page-6 recommending the use of a common encoder $D_{r}$ across all SRUs predicting the different time series components, as part of our response to a comment by reviewer #1.

5. Fixed two typos in Appendix G:
     a.  the 'batch-size' entries for NetSim and Dream-3 were mistakenly swapped in Table 7-11. We fixed it.
     b.  In table-9, we corrected the tuned value of '#Layers' hyperparameter reported for the attention-gated CNN model.

---

### Decision · Program_Chairs · 2019-12-19

**Decision:**

Accept (Poster)

**Comment:**

The authors propose a modification of the statistical recurrent unit for modelling mutliple time series and show that it can be very useful in practice for identifying granger causality when the time series are non-linearly related. The contributions are primarily conceptual and empirical. The reviewers agree that this is a useful contribution in the causality literature.